# A Clinically Relevant Classification and Staging System for Chronic Rhinosinusitis with Nasal Polyposis: A Cross-Sectional Study

**DOI:** 10.3390/diagnostics15243197

**Published:** 2025-12-14

**Authors:** Goran Latif Omer, Stefano Di Girolamo, Sahand Soran Ali, Riccardo Maurizi, Sveva Viola, Giuseppe De Donato

**Affiliations:** 1Department of Clinical Sciences, College of Medicine, University of Sulaimani, Sulaymaniyah 46001, Iraq; 2Department of Otorhinolaryngology, University of Rome Tor Vergata, 00133 Rome, Italy; stefano.di.girolamo@uniroma2.it (S.D.G.); riccardo.maurizi@ptvonline.it (R.M.); svevaviola9@libero.it (S.V.); gdedonato2@gmail.com (G.D.D.); 3College of Pharmacy, American University of Iraq—Sulaimani (AUIS), Sulaymaniyah 46001, Iraq; sahand.soran@auis.edu.krd

**Keywords:** chronic rhinosinusitis, eosinophilic CRSwNP, non-eosinophilic CRSwNP, central compartment atopic disease, phenotyping, endoscopic staging

## Abstract

**Background/Objectives:** Tissue eosinophilia plays a central role in chronic rhinosinusitis with nasal polyposis (CRSwNP), yet the spectrum of disease, particularly central compartment atopic disease (CCAD), remains underexplored. This study aimed to classify CRSwNP into three distinct phenotypes, eosinophilic CRSwNP (ECRSwNP), non-eosinophilic CRSwNP (NECRSwNP), and CCAD, based on radiologic and endoscopic features. It also proposes a novel severity-based staging system to guide clinical decision-making. **Methods:** A cross-sectional observational study was conducted in a single private clinic between January 2019 and August 2025. Patients were assessed using clinical history, paranasal sinus computed tomography (CT), and intranasal endoscopy. Key variables included symptom clusters, comorbidities, hematologic and atopy profiles, radiologic and endoscopic findings, histopathology, and pre-treatment SNOT-22 scores. **Results:** A total of 2060 patients (mean age: 29.8 ± 11 years; 51.8% male) were included. Asthma was the most frequent comorbidity (23.5%). Classification into ECRSwNP, NECRSwNP, and CCAD was achieved using integrated clinical, radiologic, and histopathologic criteria. **Conclusions:** This study presents a phenotype- and severity-based classification system for CRSwNP that incorporates endoscopic and radiologic features. This framework may enhance diagnostic accuracy and enable more tailored therapeutic strategies.

## 1. Introduction

Chronic rhinosinusitis (CRS) is an inflammatory disease of the paranasal sinuses that imposes a considerable burden on both patients and healthcare systems, affecting up to 12% of adults [1]. According to the European Position Paper on Rhinosinusitis and Nasal Polyps (EPOS 2020), the diagnosis of CRS requires 12 weeks or more of at least two symptoms, one of which must be either nasal blockage, obstruction, congestion, or discharge, with or without facial pain, pressure, and/or a reduction or loss of smell [2]. CRS can be classified phenotypically, as either CRS with nasal polyps (CRSwNP) or without (CRSsNP), or based on its underlying endotype, such as type 2 and non-type 2 inflammation [3]. Depending on disease subtype and severity, treatment approaches vary and may include medical management (corticosteroids, antibiotics, or biologics) or surgical intervention, most commonly endoscopic sinus surgery [4].

Tissue eosinophilia has been widely recognized as a major factor influencing disease severity, resistance to treatment, and postoperative recurrence in CRSwNP. As a result, most published literature has focused on eosinophilic CRSwNP (ECRSwNP), while other clinically relevant forms such as non-eosinophilic CRSwNP (NECRSwNP) and central compartment atopic disease (CCAD) have been relatively overlooked [5]. However, emerging evidence suggests that NECRSwNP and CCAD may represent distinct and diagnostically relevant phenotypes with differing inflammatory profiles, clinical behaviors, and radiologic characteristics.

NECRSwNP is primarily driven by neutrophilic inflammation, typically in response to external triggers such as microbial pathogens [6]. These cases often display a more localized distribution pattern, with dominant maxillary sinus involvement [7]. In contrast, CCAD is characterized by central involvement of the nasal cavity, including the middle turbinate, superior turbinate, and superior nasal septum. It is thought to be mediated by inhalant allergens and shares pathophysiological similarities with allergic rhinitis [8]. ECRSwNP, by comparison, presents as a more aggressive and treatment-resistant disease phenotype. It is associated with type 2 inflammation, extensive eosinophilic infiltration, and a well-documented tendency to recur following surgical or medical treatment [9].

Despite increasing interest in endotyping and subclassifying CRSwNP, the field still lacks a practical and reproducible system that allows clinicians to differentiate phenotypes and tailor treatment accordingly. Previous studies have either focused narrowly on tissue histology or have lacked comprehensive integration of endoscopic and radiologic findings. Moreover, existing classification frameworks, such as EPOS and JESREC, remain limited in their diagnostic applicability, as they primarily distinguish eosinophilic from non-eosinophilic inflammation without incorporating endoscopic origin or computed tomography (CT) patterns into a clinically useful staging system [2,10]. EPOS 2020 has proposed a refined classification of primary CRS, including diffuse CRSwNP and localized entities such as CCAD, based on endotype dominance and anatomic distribution. However, practical tools that integrate endoscopic and CT findings into a unified, phenotype specific staging system, validated in a large single cohort, are still lacking. The present study aims to operationalise these EPOS concepts by proposing an endoscopy and CT-based classification that assigns CRSwNP patients to ECRSwNP, NECRSwNP, or CCAD and by describing phenotype specific radiologic and endoscopic hallmarks in a real-world cohort of 2060 patients. These approaches also often depend on histopathological or blood biomarkers, which are not always readily available in routine practice and do not consistently guide therapeutic decision-making at the time of diagnosis. Recent work has emphasized the distinct clinical and biological characteristics of central compartment atopic disease (CCAD) and non-eosinophilic CRSwNP, yet there is still no consensus on a severity scale that links phenotype, disease burden, osteitis, and prognosis [11,12]. An integrated diagnostic system that combines endoscopic findings, CT-based severity features, and clinical correlations could therefore improve stratification, enable earlier targeted therapy, and support surgical planning. In addition, there remains no unified or validated approach for staging CRSwNP severity across these distinct phenotypes. Therefore, this study aims to address two major gaps. First, it presents a combination of endoscopic and radiologic indicators that help distinguish eosinophilic from non-eosinophilic inflammation. Second, it proposes a structured classification and staging system for CRSwNP that includes ECRSwNP, NECRSwNP, and CCAD. By correlating phenotypes with clinical, radiologic, and histopathologic data, the study provides a practical diagnostic tool to guide phenotype-specific management strategies.

## 2. Materials and Methods

### 2.1. Study Design and Setting

This is a cross-sectional observational study conducted at the first author’s private clinic in Sulaymaniyah, Iraq. All eligible data were retrospectively collected from patients diagnosed with chronic rhinosinusitis with nasal polyposis (CRSwNP) between January 2019 and August 2025. The study was reported following the Strengthening the Reporting of Observational Studies in Epidemiology (STROBE) guidelines.

### 2.2. Study Size and Participants

All adult patients (age ≥18 years) with a confirmed diagnosis of CRSwNP who had complete clinical, radiological, and endoscopic workups were included. Inclusion criteria required the availability of detailed medical history, a computed tomography (CT) scan of the paranasal sinuses, and intranasal endoscopy records. Patients with incomplete data or a history of prior sinonasal surgery were excluded. Patients with focal benign or neoplastic lesions, such as respiratory epithelial adenomatoid hamartoma (REAH), other hamartomatous or glandular lesions, inverted papilloma, sinonasal malignancy, or skull base lesions, were excluded based on endoscopic appearance, CT findings, and histopathology when indicated, in order to avoid misclassifying these conditions as CRSwNP. Patients checked for eligibility can be seen in the flow diagram in Figure 1.

### 2.3. Variables and Data Sources/Measurements

Only variables available for all patients were analyzed. These included symptomatology, comorbidities, other relevant history findings, peripheral blood tests, skin prick atopy test results, CT scan findings, intranasal endoscopy findings, tissue histopathology reports, and pre-treatment Sino-Nasal Outcome Test (SNOT-22) scores [13]. All SNOT-22 scores were self-reported by patients. All patients underwent a structured clinical evaluation including a history of allergic symptoms. Total serum IgE was measured, and specific IgE or skin prick testing was performed when available. CCAD was diagnosed in patients with typical central compartment endoscopic findings and CT patterns, interpreted in the context of an allergic background, in line with EPOS 2020, which considers CCAD a centrally localized, allergy driven form of primary CRS. The thresholds applied in this study were based on prior CRS literature and validated criteria. Total serum IgE > 100 IU/mL is a commonly used indicator of elevated IgE in sinonasal disease and allergic profiles [14,15]. Blood eosinophil cutoffs >0.24 × 10^9^/L and >0.45 × 10^9^/L were adopted from the JESREC diagnostic algorithm and subsequent CRS studies, where 0.24–0.25 × 10^9^/L reflects diagnostic eosinophilia and values >0.45 × 10^9^/L correspond to severe systemic eosinophilic inflammation associated with higher disease burden [10,16,17].

Polyp tissue was obtained during the endoscopic examination and submitted for histopathological evaluation, which was performed by a board-certified pathologist. Tissue eosinophil counts were obtained from hematoxylin- and eosin-stained paraffin sections (4 μm thick). Eosinophils were quantified under 400× magnification in five non-overlapping high-power fields (HPFs) by a board-certified pathologist blinded to clinical data. The mean eosinophil count per HPF was recorded, and tissue eosinophilia was graded as high (10–100 eosinophils per high-power field [HPF]) and very high (>100 eosinophils per HPF) [14,15,18,19,20]. These clinical and laboratory variables were used to classify CRSwNP into eosinophilic CRSwNP (ECRSwNP), non-eosinophilic CRSwNP (NECRSwNP), and central compartment atopic disease (CCAD), and to further stratify each group based on severity.

Matting of polyps was evaluated qualitatively, as no validated quantitative scoring system currently exists. Consistent with prior eosinophilic CRS literature, matting was assessed based on the degree of confluence, surface adhesion, and pseudocyst formation.

### 2.4. Endoscopy Procedure

All endoscopic examinations were performed by the first author or under direct supervision. Initially, patients underwent anterior rhinoscopy using a Killian nasal speculum and headlight. Cottonoids soaked in a solution of 2 mL xylometazoline (1 mg/mL), 2 mL lidocaine (4%), and 2 mL normal saline were placed in both nasal cavities for approximately 15 min. Patients were then placed in a supine position, the cottonoids were removed, and rigid endoscopic examination was carried out using a 4K video endoscope system (Karl Storz GmbH, Tuttlingen, Germany) with both 0° and 45° scopes. The 0° scope was used to examine the anterior nares, turbinates, and middle meatus. The 45° scope was used to visualize the axilla of the middle turbinate, olfactory cleft, and posterior middle meatus. Endoscopic findings were documented and archived as video recordings for review.

### 2.5. Bias

To minimize potential sources of bias:All eligible patients were included consecutively, with exclusion limited to those lacking essential data.Three independent specialists reviewed all CT scans and endoscopic videos to reduce inter-observer bias.Reviewers were blinded to the clinical classification of patients during image assessment.Clinical data were retrieved from electronic medical records to minimize recall bias.Confounders, such as comorbidities, were incorporated into the analysis.

### 2.6. Statistical Methods

Data were collected using Microsoft Excel (Version 16.95.1) and analyzed using R (Version 4.5.0). Descriptive statistics were performed, with categorical variables reported as frequencies and percentages and numerical variables presented as medians with interquartile ranges due to non-normal distribution. Comparative analyses between CRSwNP phenotypes (NECRSwNP, CCAD, and ECRSwNP) were conducted to assess differences in clinical and laboratory characteristics. Continuous variables (SNOT-22, serum IgE, blood eosinophils) were compared using the Kruskal–Wallis test with post hoc Dunn’s test (Bonferroni adjustment). Categorical variables (asthma, allergic rhinitis, AERD) were analyzed using the Chi-square test or Fisher’s exact test where appropriate. For stage-based comparisons, the Jonckheere–Terpstra test was applied to continuous variables and the Cochran–Armitage trend test to categorical variables. Statistical significance was defined as *p* < 0.05. Inter-observer agreement was evaluated using the Fleiss Kappa statistic via the ‘irr’ package in R, with a Kappa value above 0.75 interpreted as strong agreement.

### 2.7. Ethical Considerations

This study was approved by the Ethical Committee of the University of Sulaimani, Approval Report Number 158, at its 15th meeting on 17 August 2025. All procedures were conducted in accordance with the ethical standards of the Declaration of Helsinki. Data are available upon reasonable request from the corresponding author.

## 3. Results

### 3.1. General Findings

A total of 2060 patients were included. The mean age was 29.8 ± 11 years, and 52.2% were male. Comprehensive demographic data, comorbidities, and laboratory findings are summarized in Table 1.

### 3.2. Comparative Analysis Among Phenotypes

Statistical comparisons were performed to evaluate inter-group differences across the three CRSwNP phenotypes. The median SNOT-22 score differed significantly among NECRSwNP, CCAD, and ECRSwNP (41 [IQR 20.5], 43 [IQR 22], and 69 [IQR 31], respectively; Kruskal–Wallis *p* < 0.001). Serum IgE and blood eosinophil levels also showed significant variation between groups (*p* < 0.001 for both). Regarding comorbidities, asthma prevalence was highest in ECRSwNP (36%), intermediate in CCAD (18.8%), and lowest in NECRSwNP (5.7%), while allergic rhinitis was observed in all CCAD cases (100%) and in 32% of ECRSwNP patients, but absent in NECRSwNP (*p* < 0.001). Aspirin-exacerbated respiratory disease (AERD) was confined to the ECRSwNP group (7%). These differences collectively confirm that the proposed phenotypes exhibit distinct clinical, immunologic, and radiologic profiles.

### 3.3. Classification of CRSwNP

This study established a clinically applicable classification and staging system for chronic rhinosinusitis with nasal polyposis (CRSwNP). Subgroups were defined through integration of histopathological, hematological, immunological, endoscopic, and radiologic characteristics. The proposed system comprises three major phenotypes:Non-Eosinophilic CRSwNP (NECRSwNP)Central Compartment Atopic Disease (CCAD)Eosinophilic CRSwNP (ECRSwNP)

The criteria and staging characteristics for each group are detailed in Table 2.

### 3.4. Subtype Profiles

NECRSwNP: This phenotype accounted for 476 patients (23.1%), predominantly female (60.1%), with a median age of 41 years (IQR 7). Patients presented mainly with nasal obstruction and postnasal drip, often without atopy. Asthma was reported in 5.7%, while serum IgE and blood eosinophil levels were generally low. Histopathology confirmed neutrophilic infiltration without tissue eosinophilia. Radiologically, disease was limited, showing incomplete sinus opacification, with a median GOSS of 3 (IQR 1) and E:M ratio of 0.75 (IQR 0.30). The median SNOT-22 score was 41 (IQR 20.5).CCAD: There were 500 patients (24.3%), with near-equal sex distribution (51% male) and a median age of 24 years (IQR 5). All met the criteria for allergic rhinitis and had positive skin-prick tests (perennial allergens only). Asthma occurred in 18.8%. Histology demonstrated marked tissue eosinophilia (≥10 per HPF) despite low peripheral eosinophil counts. CT imaging typically showed central-compartment predominance (“black halo sign”), with sparing of lateral sinus walls and no pansinusitis or neo-osteogenesis. The median GOSS was 5 (IQR 4) and E:M ratio was 2.00 (IQR 0.40), consistent with localized allergic inflammation. The median SNOT-22 score was 43 (IQR 22).ECRSwNP: This was the most frequent subtype (1084 patients, 52.6%), mostly male (58.1%), with a median age of 32 years (IQR 6). Symptoms included nasal obstruction and anosmia ± snoring. Asthma was present in 36%, AERD in 7%, and allergic rhinitis in 32%. Nearly all patients had elevated serum IgE, positive multi-allergen skin-prick tests, and blood eosinophilia (>0.24 × 10^9^/L). Histopathology confirmed tissue eosinophilia >10 per HPF, frequently >100 per HPF in advanced cases. Radiologically, the disease was more extensive, with diffuse sinus opacification, osteitis, and occasionally neo-osteogenesis. The median GOSS was 25 (IQR 16) and E:M ratio 1.20 (IQR 0.40), indicating high ethmoidal involvement relative to maxillary disease. The median SNOT-22 score was 69 (IQR 31).

All characteristic endoscopic and CT features defining each phenotype and stage are provided in Table 2 and Figure 2, Figure 3, Figure 4 and Figure 5.

### 3.5. Inter-Observer Agreement

To assess inter-observer reliability in the diagnosis and staging of CRSwNP subtypes, Fleiss’ Kappa analyses were performed. Prior to formal staging, three independent observers reviewed a set of 500 randomly selected cases (including endoscopic videos and CT scans) to classify the CRSwNP subtype. From these, 50 cases were randomly selected to evaluate inter-observer agreement in subtype diagnosis, yielding a Kappa value of 0.899 (95% CI, 0.861–0.937; z = 15.5; *p* < 0.01), indicating almost perfect agreement. Subsequently, 150 cases (50 CCAD, 50 ECRSwNP, and 50 NECRSwNP) with consensus on subtype were used to assess inter-observer agreement in staging. The Fleiss’ Kappa values (with 95% confidence intervals) were as follows:•ECRSwNP: κ = 0.938 (95% CI, 0.905–0.971; z = 16.0; *p* < 0.01);•NECRSwNP: κ = 0.946 (95% CI, 0.912–0.980; z = 11.6; *p* < 0.01);•CCAD: κ = 0.891 (95% CI, 0.845–0.937; z = 10.9; *p* < 0.01).

All results demonstrated almost perfect agreement (κ > 0.81) and were statistically significant (*p* < 0.05), reinforcing the consistency and reproducibility of the proposed classification system.

## 4. Discussion

### 4.1. General Evaluation

Chronic rhinosinusitis with nasal polyposis (CRSwNP) has been classified in various ways, including by phenotype and endotype. However, a clinically practical and therapeutically informative classification that incorporates both diagnostic and prognostic dimensions remains lacking. This study proposes a novel system that categorizes CRSwNP into three clinically distinct groups: ECRSwNP, NECRSwNP, and CCAD. These categories were selected based on differences in pathophysiology, clinical presentation, and treatment response, aligning with findings by Grayson et al. [15]. Recognizing the internal heterogeneity within each group, a staging system was also developed to reflect disease severity. While many studies have described individual clinical, endoscopic, or radiologic characteristics of these subtypes, the current study provides an integrated framework that consolidates and builds upon these contributions. The proposed classification and staging system was primarily derived from the authors’ own large single-center cohort, supported by integration of concepts described in international literature. Rather than reproducing existing models such as JESREC or Lee et al., which focus mainly on diagnostic differentiation, the present system was developed to bridge diagnostic and prognostic dimensions through combined radiologic, endoscopic, and histopathologic parameters. This approach reflects the cumulative clinical experience of the authors’ multidisciplinary research group, refined through validation against prior global evidence and adapted for practical use in routine endoscopic evaluation. These findings should also be interpreted in the context of existing classification systems. Traditional approaches such as the JESREC score and other histopathology-based methods have significantly advanced understanding of disease mechanisms but remain limited in clinical applicability, as they do not incorporate endoscopic origin or radiological severity markers into their diagnostic frameworks [10]. Similarly, EPOS 2020 recommendations focus on eosinophilic versus non-eosinophilic inflammation without providing a structured severity scale that links phenotype to disease burden, osteitis, or prognosis [2]. Recent studies have highlighted the clinical relevance of central compartment atopic disease (CCAD) and non-eosinophilic phenotypes, both of which present distinct inflammatory characteristics and treatment responses [11,12]. By integrating polyp origin, CT-based osteitis evaluation, and severity staging, the present study addresses these limitations and proposes a practical framework that can be applied during routine diagnostic assessment. This approach may allow clinicians to stratify patients more accurately, guide decisions on surgical extent and biologic therapy, and better predict long-term outcomes.

### 4.2. Radiologic and Endoscopic Signs of Eosinophilia

Several studies have described endoscopic and radiologic features associated with eosinophilic inflammation in nasal polyps. This study systematically evaluated those characteristics to establish a non-invasive, clinically applicable method for identifying eosinophilic nasal polyps (ENP) without relying on laboratory or histopathologic tests.

#### 4.2.1. Endoscopic Features

•Polyps from extraordinary areas: Polyps originating from uncommon sites such as the olfactory cleft, axilla of the middle turbinate, lacrimal crest, and nasal septum were predominantly observed in ENP [21,22].•Matting of polyps: A hallmark of ENP, matting results from fibrin deposition, extracellular matrix remodeling, and impaired fibrinolysis, all mediated by eosinophil-derived proteins [23,24,25,26].•Extensive mucosal edema and viscous secretions: These are linked to Th2-mediated inflammation (IL-4, IL-5, IL-13), which promotes goblet cell hyperplasia and epithelial damage via major basic protein and eosinophil cationic protein [25,27,28].

#### 4.2.2. Radiological (CT) Features

•Bony changes: Osteitis, ethmoid expansion, and neo-osteogenesis were common in ENP and are thought to be mediated by TGF-β1-induced ALP upregulation [29,30,31,32].•Central compartment opacity: Frequently referred to as the black halo sign, this finding, characterized by central mucosal thickening with peripheral sinus sparing, is common in both CCAD and ENP [33].

According to the findings discussed above, many features of ENPs are shared between ECRSwNP and CCAD, while they are largely absent in NECRSwNP.

### 4.3. Clinical Features and Patient Demographics

Previous studies have shown that CCAD patients tend to be the youngest, followed by those with ECRSwNP, while NECRSwNP patients are generally older. Although all three groups share typical CRS symptoms, CCAD patients are more likely to exhibit allergic rhinitis symptoms, ECRSwNP patients frequently report anosmia, and NECRSwNP patients often present with persistent coughing. In terms of laboratory findings, total IgE levels are typically highest in CCAD, moderately elevated in ECRSwNP, and rarely elevated in NECRSwNP. Blood eosinophilia (≥0.24 × 10^9^ cells/L) has been reported almost exclusively in ECRSwNP [7,15,34,35]. These trends were also observed in this study’s patients, although the prevalence of asthma (23.5%) and AERD (3.7%) was notably lower compared to earlier reports, where asthma affected 69–74% and AERD affected 24–27% of ECRSwNP patients [15,36]. This discrepancy may be attributed to regional or population-based differences.

### 4.4. Endoscopic and Radiologic Staging Insights

Regarding CCAD, a recent review described central compartment polyp origins, particularly from the middle turbinate, with more severe disease involving the posterior superior septum and superior turbinate, findings consistent with Stage 2 CCAD in this study [8]. The black halo sign, reflecting central mucosal thickening with peripheral sparing, was consistently observed. Additionally, this study identified the inferior turbinate as a potential marker of severity, with Stage 2 CCAD showing a mulberry appearance and posterior turbinate hypertrophy. Osteitis, measured using the Global Osteitis Scoring Scale (GOSS), was insignificant (0–5) in Stage 1 and mild (<20) in Stage 2 [31,37]. Although CCAD is classically confined to the middle turbinate, superior turbinate, and posterosuperior nasal septum, recent evidence suggests that advanced disease may extend inferiorly. In Stage 2 CCAD, we observed a characteristic ‘mulberry appearance’ of the posterior inferior turbinate. This finding is likely secondary rather than primary and can be explained by several mechanisms. First, Th2-mediated inflammation can extend beyond the central compartment in more severe disease, with inferior turbinate papillary remodeling reported in recent CCAD and allergic CRS studies [11,12]. Second, the inferior turbinate is highly vascular and susceptible to dependent edema and venous congestion driven by eosinophil-derived mediators [24,25]. Third, enlargement of the central compartment narrows airflow pathways, causing turbulent redirection toward the inferior turbinate, which promotes posterior hypertrophy, a mechanism also noted in CCAD airflow analyses [8,19]. Taken together, these findings support our interpretation that inferior turbinate involvement represents a severity marker in CCAD rather than a primary site of disease origin.

Findings associated with tissue eosinophilia, such as matted polyps, polyps from the lacrimal crest or olfactory cleft, and neo-osteogenesis, were predominantly observed in ECRSwNP. These features guided the staging system in this study. For instance, polyps from the lateral surface of the middle turbinate were categorized as Stage 1; the axilla of the middle turbinate and lacrimal crest defined Stage 2; and olfactory cleft involvement indicated Stage 3. On CT, the basal lamella served as a key landmark: Stage 1 showed pre-basal lamella opacity, Stage 2 involved the basal lamella, and Stage 3 exhibited pre-, peri-, and post-basal lamella opacity. Existing scoring systems such as JESREC and that of Lee et al. are primarily diagnostic tools and do not include staging, limiting their utility for severity stratification [10,38].

CCAD was distinguished from ECRSwNP based on several clinical and pathophysiological features that support its recognition as an independent subtype within the CRSwNP spectrum rather than an early or localized form of ECRSwNP. Although both share certain eosinophilic characteristics, CCAD is confined to the central compartment, particularly the middle turbinate, septum, and superior turbinate, and lacks pansinusitis, neo-osteogenesis, and significant osteitis. While tissue eosinophilia is present, CCAD is uniquely associated with allergic rhinitis and demonstrates normal blood eosinophil counts and IgE levels restricted to perennial allergens. In contrast, ECRSwNP exhibits systemic eosinophilic inflammation, elevated serum IgE, multisinus involvement, and allergy to multiple allergens. The inclusion of CCAD within the CRSwNP classification is therefore justified by its overlapping inflammatory mechanisms and polypoid nature, while its distinct anatomical localization and allergic profile warrant recognition as a separate clinical subtype. These findings align with recent literature describing CCAD as a localized, allergy-driven manifestation of Th2 inflammation within the sinonasal tract.

NECRSwNP was diagnosed based on exclusion of hallmark features seen in ECRSwNP and CCAD, along with identification of distinct characteristics, namely, simple polyp morphology, origins from ordinary anatomic areas, and predominant maxillary sinus involvement. Staging in NECRSwNP was based on the extent of nasal cavity obstruction and whether CT scans showed isolated maxillary involvement or full pansinusitis. From a therapeutic perspective, our classification is aligned with the current EPOS 2020 recommendations. All three phenotypes (ECRSwNP, NECRSwNP, and CCAD) are managed with saline irrigation and intranasal corticosteroids as baseline therapy, which remain the cornerstone of medical management in CRS. Systemic corticosteroids and biologic agents targeting type 2 inflammation are currently reserved for patients with severe, uncontrolled type 2 dominant disease, which in our framework corresponds primarily to ECRSwNP with high tissue and blood eosinophilia and extensive polyposis. In contrast, NECRSwNP and CCAD, which display non-type 2 or predominantly localized allergic inflammation, are less likely to meet current indications for biologic therapy. Although our study was not designed to compare treatment responses between phenotypes, the proposed classification provides a pragmatic structure for future prospective trials evaluating differential outcomes of intranasal corticosteroids, surgery, and biologics across these three entities.

### 4.5. Study Limitations and Future Directions

EPOS 2020 has delineated primary CRS phenotypes, including eCRS, CRSwNP, and CCAD, within a conceptual framework based on endotype and anatomic distribution. Our contribution is to translate these concepts into a unified, operational classification and staging system based on routine endoscopy and CT, validated in a large cohort and enriched with phenotype specific features such as matting, central ‘halo’ patterns, osteitis burden, and inferior turbinate involvement in advanced CCAD. This approach offers a practical tool that can be applied in daily practice, including in settings where extensive biomarker testing is not feasible, and provides a structured platform for future outcome studies. Although this was a single-center study, it was conducted at a high-volume tertiary referral center serving a broad and diverse population. This enhanced the generalizability of the proposed classification. Nevertheless, multicenter validation remains essential. Future studies in varied clinical settings are needed to confirm the reproducibility and international applicability of this system. From a clinical perspective, the proposed classification and staging system has the potential to transform how CRSwNP is evaluated and managed in daily practice. By integrating endoscopic findings, CT-based severity markers, and histopathologic correlations, this approach enables more accurate phenotype identification and facilitates early treatment decisions tailored to disease type and severity. It may also support more precise selection of surgical strategies and biologic therapies, improving outcomes while reducing unnecessary interventions. Ultimately, widespread adoption of this framework could standardize diagnostic workflows and enhance the personalization of care for patients with CRSwNP.

## 5. Conclusions

This study presents a clinically applicable classification and staging system for chronic rhinosinusitis with nasal polyposis (CRSwNP) that integrates endoscopic, radiologic, and histopathologic parameters. By differentiating eosinophilic CRSwNP (ECRSwNP), non-eosinophilic CRSwNP (NECRSwNP), and central compartment atopic disease (CCAD), the proposed framework captures disease heterogeneity and provides a practical tool to improve diagnostic precision. Incorporating severity staging strengthens its clinical relevance by linking phenotype to disease burden, treatment response, and prognosis. Implementation of this system in daily practice could support earlier, more personalized interventions, guide the extent of surgical management, and inform biologic therapy decisions. Future multicenter studies are needed to validate this approach across diverse populations and clinical settings.

## Figures and Tables

**Figure 1 diagnostics-15-03197-f001:**
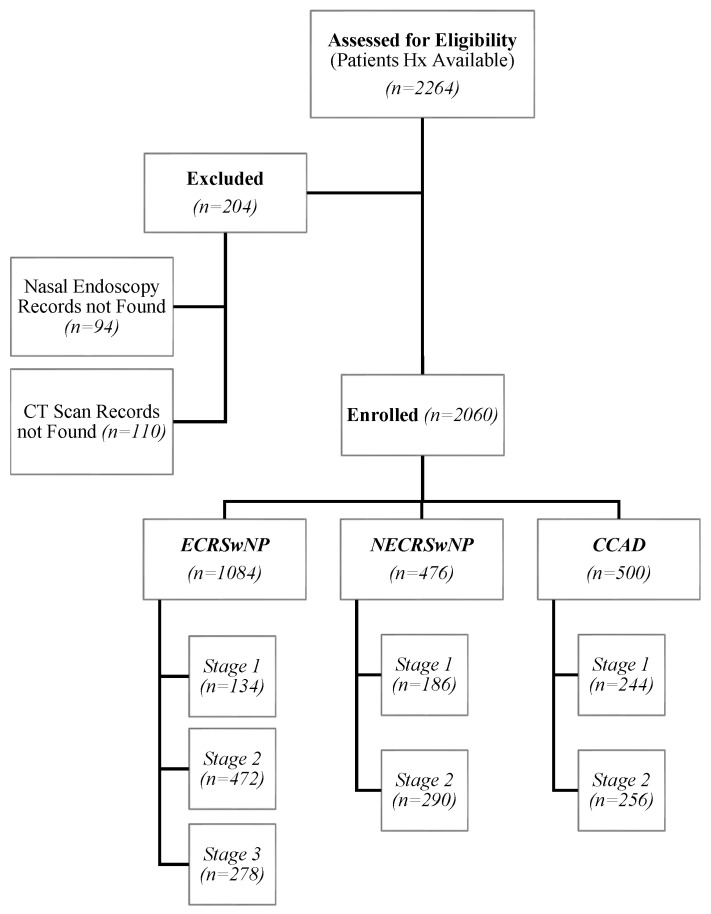
Study flow and phenotypic allocation. Eligibility, exclusions, final cohort (*n* = 2060), and allocation to NECRSwNP, CCAD, and ECRSwNP with stage counts.

**Figure 2 diagnostics-15-03197-f002:**
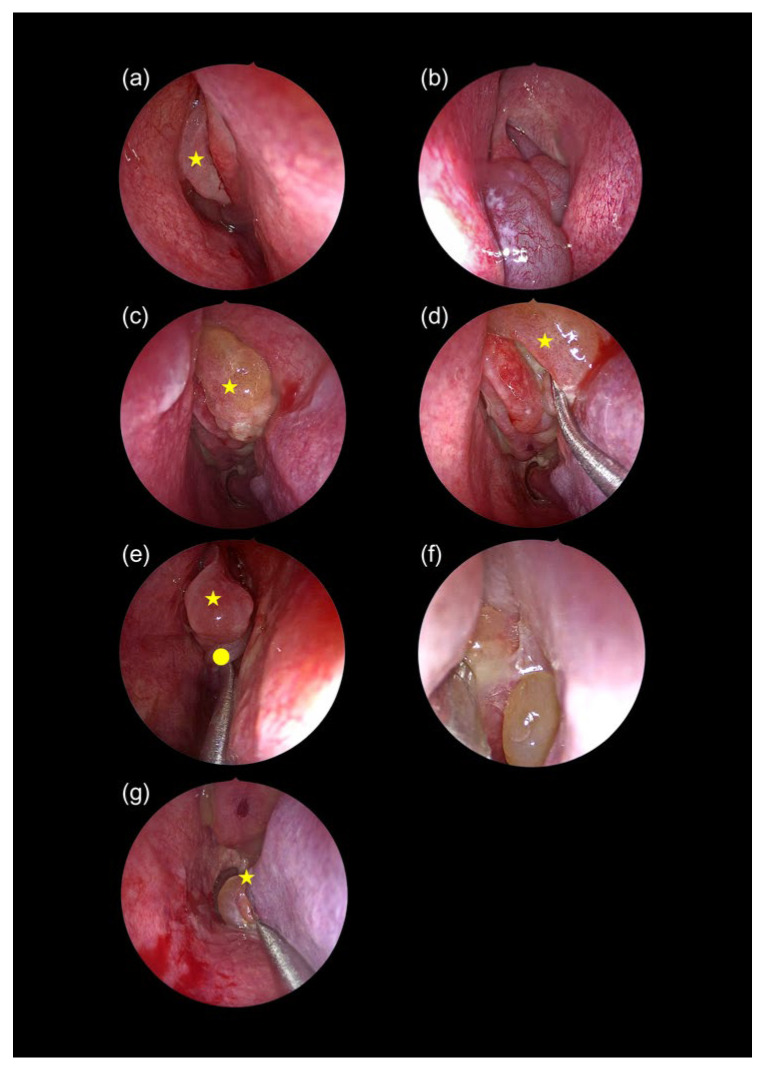
Endoscopic features distinguishing NECRSwNP and CCAD. NECRSwNP: (**a**) single polyp (star) from a normal anterior ethmoidal site; (**b**) simple pedunculated polyps filling the nasal cavity with discolored discharge. CCAD: (**c**) polyp from the middle meatus (star); (**d**) polyp from the uncinate process (star) and edema of the middle turbinate tip; (**e**) edematous middle turbinate (star) and polyp from its medial surface (circle); (**f**) tenacious secretions; (**g**) mulberry appearance of the posterior inferior turbinate (star).

**Figure 3 diagnostics-15-03197-f003:**
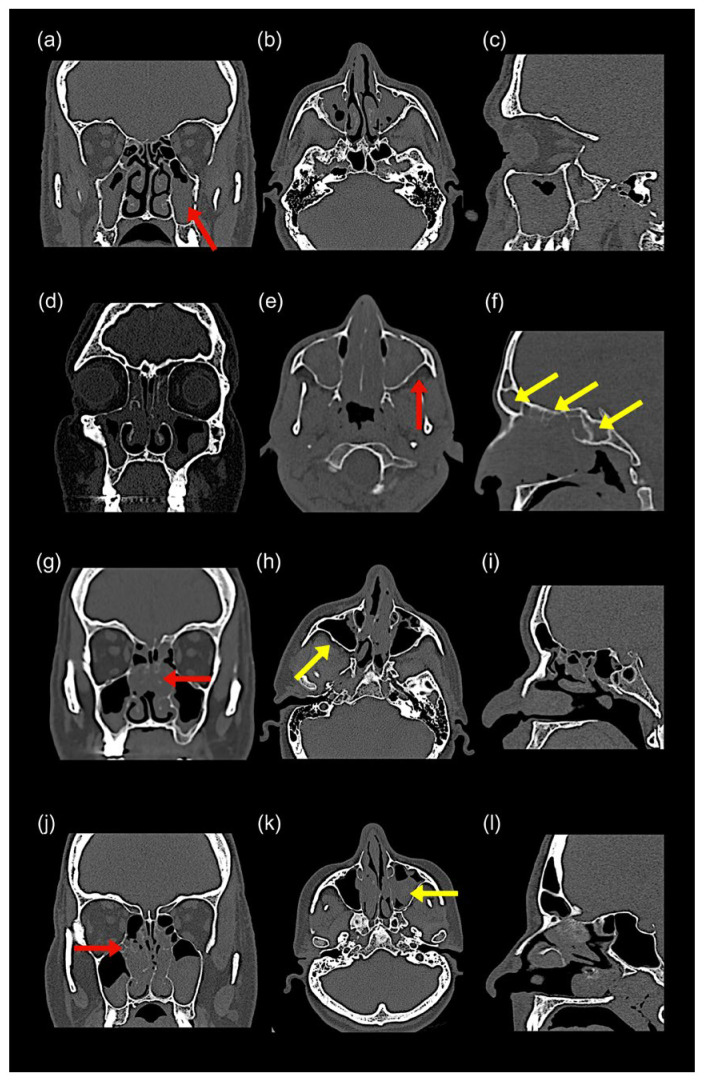
CT characteristics of NECRSwNP and CCAD. CT findings across coronal, axial, and sagittal planes. NECRSwNP: Stage 1 (**a**–**c**) shows low E:M ratio with partial maxillary opacification and no bony expansion (red arrow); Stage 2 (**d**–**f**) shows pansinusitis (yellow arrows) with mild maxillary expansion (red arrow). CCAD: Stage 1 (**g**–**i**) shows the black halo sign (red arrow) with sparing of lateral sinus walls (yellow arrow); Stage 2 (**j**–**l**) shows peripheral opacities from outflow obstruction (yellow arrow) and mild osteitis (red arrow), without neo-osteogenesis.

**Figure 4 diagnostics-15-03197-f004:**
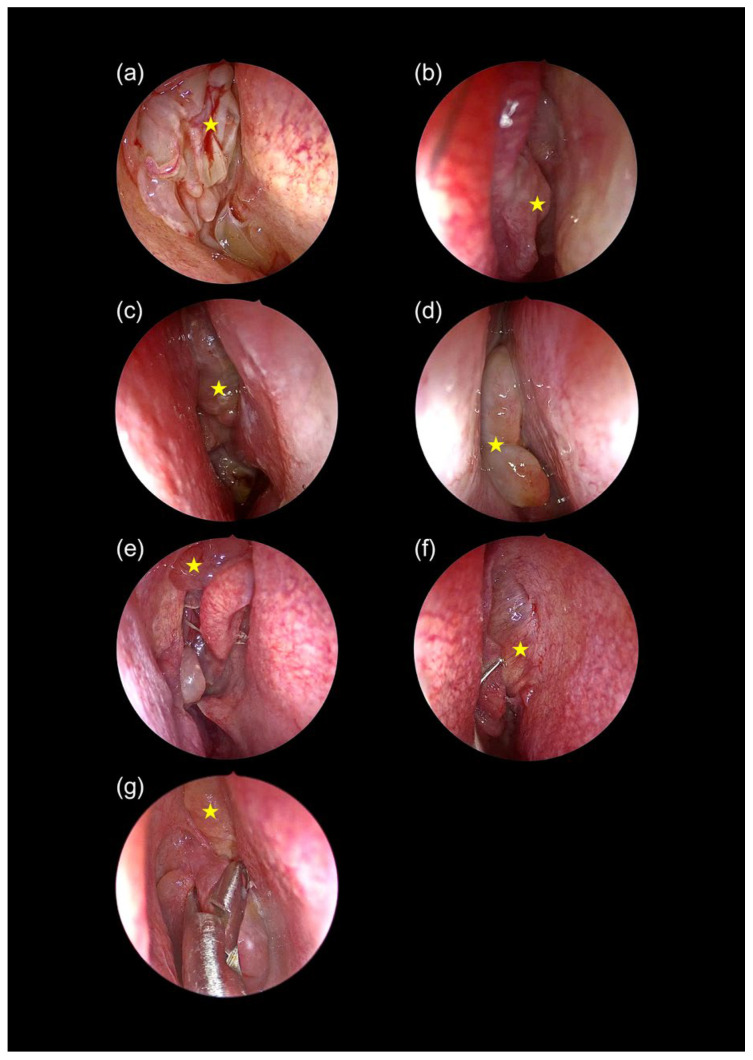
Endoscopic findings in ECRSwNP. (**a**) Multiple small polyps (star) matted together in the OMC; (**b**) polyp (star) from the lateral surface of the middle turbinate; (**c**,**d**) large confluent polyps (star) filling the nasal cavity; (**e**) polyp (star) from the axilla of the middle turbinate; (**f**) polyp (star) from the lacrimal crest; (**g**) polyp (star) from the olfactory cleft.

**Figure 5 diagnostics-15-03197-f005:**
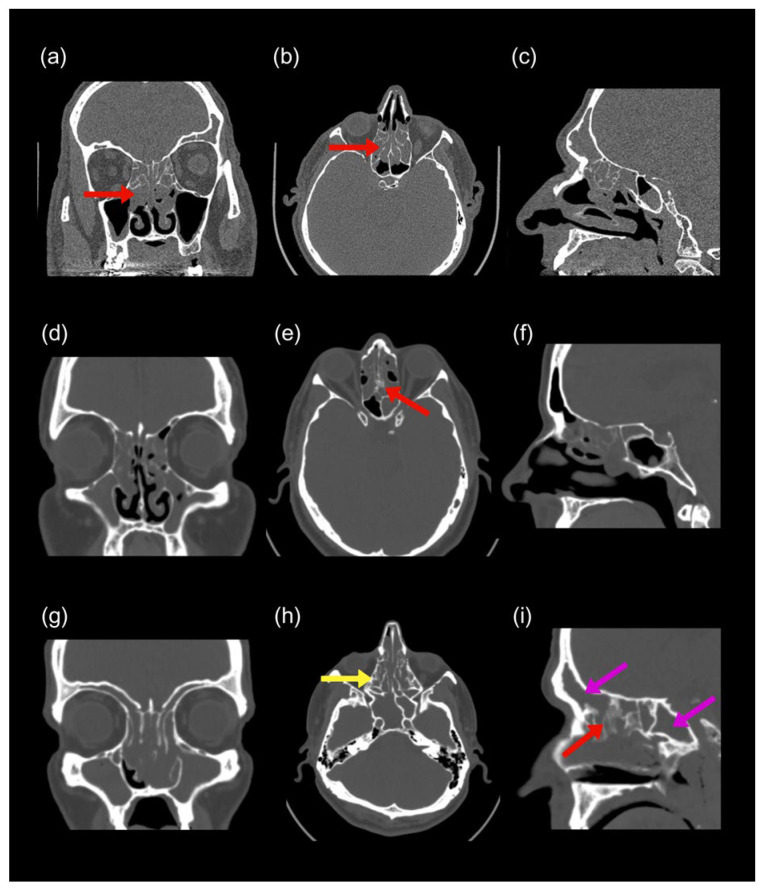
CT features of ECRSwNP across stages. Coronal, axial, and sagittal CT images. Stage 1 (**a**–**c**): diffuse mucosal thickening, ethmoid-predominant opacity (high E:M; red arrows). Stage 2 (**d**–**f**): partial opacification of all sinuses and focal posterior ethmoid osteitis (red arrow). Stage 3 (**g**–**i**): pansinusitis (pink arrows) with diffuse osteitis (yellow arrow), neo-osteogenesis (red arrow), and sinus expansion.

**Table 1 diagnostics-15-03197-t001:** Baseline characteristics of patients with CRSwNP subtypes.

Variables		NECRSwNP	CCAD	ECRSwNP	Total
Age (Year [IQR])		41 (IQR 7)	24 (IQR 5)	32 (IQR 6)	29.8 ± 11
Sex	Male	190 (39.9%)	255 (51%)	630 (58.1%)	1075 (52.2%)
	Female	286 (60.1%)	245 (49%)	454 (41.9%)	985 (47.8%)
Presenting Complaint	Nasal Obstruction, **Anosmia** ± Snoring	0	0	1084 (100%)	1084 (52.6%)
	Nasal Obstruction, **Postnasal Drip** ± Cough	476 (100%)	0	0	476 (23.1%)
	Nasal Obstruction, Rhinorrhea, and **Sneezing**	0	500 (100%)	0	500 (24.3%)
Associated Disease	**Asthma**	27 (5.7%)	94 (18.8%)	390 (36%)	511 (24.8%)
	**Aspirin-Exacerbated Respiratory** **Disease**	0	0	76 (7%)	76 (3.7%)
	**Allergic Rhinitis**	0	500 (100%)	346 (32%)	846 (41.1%)
Total Serum IgE Count	More than **100 IU/mL**	27 (5.7%)	0	1035 (95.5%)	1062 (51.6%)
	**100 IU/mL** or less	449 (94.3%)	500 (100%)	49 (4.5%)	998 (48.4%)
Blood Eosinophils	Less than or equal to **0.24 × 10^9^** cells/L	476 (100%)	500 (100%)	0	976 (47.4%)
	More than **0.24 × 10^9^** cells/L	0	0	418 (38.6%)	418 (20.3%)
	More than **0.45 × 10^9^** cells/L	0	0	666 (61.4%)	666 (32.3%)
Skin Prick Test	Positive test (perennial allergen only)	0	500 (100%)	0	500 (24.3%)
	Positive test (multiple allergens/nonspecific)	0	0	1084 (100%)	1084 (52.6%)
Tissue Histopathology	Neutrophil Infiltration	476 (100%)	0	0	476 (23.1%)
	Eosinophil Count **10–100** per HPF	0	437 (76.4%)	418 (38.6%)	855 (41.5%)
	Eosinophil Count **>100 per** HPF	0	63 (24.6%)	666 (61.4%)	729 (35.4%)
**SNOT-22** ^§^ *(Median [IQR])*		41 (IQR 20.5)	43 (IQR 22)	69 (IQR 31)	42.5 (IQR 35)
**GOSS** * *(Median [IQR])*		3 (1)	5 (4)	25 (16)	12 (9)
**E:M Ratio** ^†^ *(Median [IQR])*		0.75 (0.30)	2.00 (0.40)	1.20 (0.40)	1.20 (0.50)
Total		476 (23.1%)	500 (24.3%)	1084 (52.6%)	2060 (100%)

^§^ SNOT-22 = Sino-Nasal Outcome Test-22; * GOSS = Global Osteitis Scoring Scale; ^†^ E:M = Ethmoid-to-Maxillary opacity ratio.

**Table 2 diagnostics-15-03197-t002:** Types of Chronic Rhinosinusitis with Nasal Polyposis and their Stages.

Group 1: NECRSwNP (Non-Eosinophilic Chronic Rhinosinusitis with Nasal Polyposis)			
Stages	Endoscopy Findings	CT Findings	Notes
Stage 1	• Simple pedunculated polyps from the middle meatus extend to the nasal cavity and can be separated from each other and from their origin. Polyps do not arise from extraordinary area. • Normal nasal mucosa • Thick, discolored discharge around the polyp	Incomplete opacification of the sinuses, especially the maxillary sinus (low E:M score ^†^)	• GOSS * = non-significant (<5) • No neo-osteogenesis
Stage 2	• Stage 1 Endoscopy Findings + • Polyps filling in the whole nasal cavity	• Pansinusitis (complete opacification of all sinuses) • Mild maxillary sinus expansion	
**Group 2: CCAD (** * **Central Compartment Atopic Disease** * **)**			
**Stages**	**Endoscopy Findings**	**CT Findings**	**Notes**
Stage 1	• Polyp/s from: middle meatus, anterior and medial surfaces of middle turbinate, corresponding areas of septum and uncinate process • Edema of the middle turbinate tip and its medial surface • Osteomeatal complex edema and polyp • Tenacious secretion	• Black Halo Sign: central thickening of the turbinates and septum with near-normal peripheral sinus mucosa • Sparing of the roof and lateral walls of the sinus cavity • GOSS = non-significant (<5)	• No pansinusitis • No sinus expansion nor neo-osteogenesis
Stage 2	• Stage 1 Endoscopy Findings + • Edema of the superior turbinate and posterior nasal septum • Mulberry appearance of the inferior turbinate	• Stage 1 CT Findings + • Lateral sinus opacity due to obstruction of outflow tracts • GOSS = non-significant or mild (<20) • Hypertrophy of the posterior end of the inferior turbinate	
**Group 3: ECRSwNP (** * **Eosinophilic Chronic Rhinosinusitis with Nasal Polyposis** * **)**			
**Stages**	**Endoscopy Findings**	**CT Findings**	
Stage 1	Small multiple polyps matted together, from: Middle meatus, lateral surface of middle turbinate, and/or may extend to the nasal cavity	• Pre-basal lamellae opacity • Diffuse sinus wall mucosa thickening and bony changes • GOSS = mild (5–20)	
Stage 2	• Stage 1 Endoscopy Findings + • Multiple huge polyps matted together, from: Axilla of the middle turbinate, both medial and lateral surfaces of the middle turbinate, and/or lacrimal crest	• Stage 1 CT Findings + • Opacity of middle meatus, anterior ethmoid cells, posterior ethmoid cells, maxillary sinus, and partial opacity of frontal and sphenoid • Calcification or partial bony wall thickening, especially in the anterior and posterior ethmoid cells’ walls • GOSS = moderate (20–35)	
Stage 3	• Stage 2 Endoscopy Findings + • Polyps from the olfactory clef ^§^	• Stage 2 CT Findings + • Pre- and post-basal lamellae opacity • Expansion of all sinuses • Neo-osteogenesis • Pansinusitis (complete opacification of all sinuses) • GOSS = severe (higher than 35)	

^†^ Ethmoid-to-maxillary (E:M) opacification ratio. * Global Osteitis Scoring System. ^§^ The severity of olfactory cleft polyposis can also be explained by the endoscopic olfactory cleft score and radiologic olfactory cleft grade.

## Data Availability

The datasets used and/or analyzed during the current study are available from the corresponding author on reasonable request.

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
