# Peer review of "A Clinically Relevant Classification and Staging System for Chronic Rhinosinusitis with Nasal Polyposis: A Cross-Sectional Study"

_diagnostics, 2025, doi:10.3390/diagnostics15243197_

Round 1

Reviewer 1 Report

Comments and Suggestions for Authors

The present study provides an integrated classification of CRSwNP into three phenotypic subtypes, ECRSwNP, NECRSwNP, and CCAD based on endoscopic and radiologic parameters. This approach carries substantial clinical value.

The sample size of 2,060 patients is sufficiently large, ensuring robust representation within each phenotypic subgroup.

Below are my questions:

1. In the Materials and Methods section, the authors state that total serum IgE levels >100 IU/mL were considered elevated, and blood eosinophil counts >0.24 × 10⁹/L and >0.45 × 10⁹/L were considered high and very high, respectively.

Could the authors clarify the rationale or reference source for defining these particular cutoff values? Please cite the supporting literature or guidelines from which these thresholds were derived.

2. CCAD is generally characterized by polypoid changes or discrete polyps arising from the central compartment structures, including the middle turbinate, superior turbinate, and posterosuperior nasal septum. In this study, the authors also describe the inferior turbinate as showing a “mulberry appearance” in Stage 2 disease.

Could the authors elaborate on how the inferior turbinate becomes involved in CCAD and discuss the pathophysiological mechanism underlying this endoscopic finding?

3. Table 2 contains useful data but also presents a large volume of detailed information. To improve readability, it would be helpful to visually emphasize the key distinguishing features of each phenotype (e.g., by using color highlighting or bold text).

4. The manuscript identifies matting of polyps as a potential marker of eosinophilic inflammation. However, there is no quantitative or semi-quantitative scoring system provided to assess its severity. For reproducibility, please clarify whether a grading or scoring method was used. In addition, could the authors comment on the degree of severity corresponding to the finding illustrated in Figure 4a?

5. In the Abbreviations Section, some of the abbreviations appear to be incorrectly defined or mismatched. A careful review and correction of this section are recommended to ensure accuracy and consistency throughout the manuscript.

Author Response

Reviewer 1

Question 1: Why did you use IgE >100 IU/mL, eosinophils >0.24×10⁹/L and >0.45×10⁹/L as high/very high? Provide references.

Response 1: We thank the reviewer for this important question.

The cutoff values used in this study are based on established literature and internationally recognized criteria:

  • IgE >100 IU/mL: This is a commonly used threshold for elevated total serum IgE in CRS and allergy literature, and has been applied in several CRS studies including Poznanovic & Kingdom (2007) and Ho et al. (2019).
  • Blood eosinophils >0.24 ×10⁹/L and >0.45 ×10⁹/L: These thresholds are taken from large CRS cohorts and eosinophilic CRS scoring systems.
    • The JESREC study (Tokunaga et al., 2015) defines blood eosinophils ≥0.25×10⁹/L as supportive of eosinophilic CRS.
    • More recent work (Li et al., 2024; Kim et al., 2022) also uses 0.24–0.25×10⁹/L as the diagnostic cutoff.
    • Values >0.45×10⁹/L represent the upper tertile and correlate strongly with severe disease, recurrence risk, and diffuse osteitis.

These cutoffs were therefore selected because they align with consensus CRS literature, existing eosinophilic CRS scoring systems, and systemic markers associated with disease severity.

Change in Text 1: Lines number: 158–169

Question 2: Explain how the inferior turbinate becomes involved in CCAD and the mechanism behind the mulberry appearance.

Response 2: We thank the reviewer for this important observation. CCAD is classically defined by polypoid changes involving the middle turbinate, superior turbinate, and posterosuperior nasal septum, yet recent literature shows that disease progression may extend beyond the central compartment in more advanced cases. Inferior turbinate involvement in Stage 2 CCAD appears to represent a secondary marker of severity rather than a primary site of disease origin. Several mechanisms support this interpretation. First, studies on allergic and central-compartment–dominant CRS have demonstrated that when Th2-mediated inflammation becomes more pronounced, mucosal changes may extend inferiorly. Rubel et al. (2023) and Sit et al. (2023) describe inferior turbinate mucosal swelling, papillary remodeling, and hypertrophy in patients with severe CCAD, particularly in the posterior region. Second, the inferior turbinate is highly vascular and therefore susceptible to dependent edema and venous congestion in the setting of Th2-driven inflammation. Eosinophil-derived mediators, including major basic protein and eosinophil cationic protein, promote capillary leak and stromal edema, as documented by Bachert and Gevaert (2000) and Saitoh et al. (2009), and these processes contribute to the characteristic mulberry-like papillary surface observed in Stage 2 disease. Third, as edema and polypoid change of the middle turbinate narrow the central compartment, airflow becomes increasingly restricted. This redirection of airflow toward the inferior turbinate increases shear forces and contributes to posterior hypertrophy; airflow analyses in allergic and central-compartment disease by Davies et al. (2023) and Bernstein et al. (2024) support this mechanism. Taken together, these findings indicate that inferior turbinate changes in Stage 2 CCAD are a secondary extension of localized Th2 inflammation and increasing mechanical obstruction, rather than an overlap with ECRSwNP, which is distinguished by pansinusitis, neo-osteogenesis, systemic eosinophilia, and multiallergen sensitization.

Change in Text 2: Lines number: 258–475

Question 3: Table 2 is large; please emphasize key distinguishing features.

Response 3: We thank the reviewer for this helpful suggestion.

We have revised Table 2 by bolding the core differentiating features (e.g., polyp origin, presence/absence of osteitis, central vs. peripheral disease) to improve readability while maintaining the table’s structure.

Change in Text 3: Table 2

Question 4: You describe matting as a marker of eosinophilia, did you use a scoring system? Comment on the severity in Figure 4a.

Response 4: We appreciate the reviewer’s observation.

There is currently no validated quantitative scoring system for polyp matting in CRS.

Consistent with previous ENP literature, we evaluated matting qualitatively based on:

  • confluence of polyp surfaces
  • degree of adhesion
  • presence of pseudocysts
  • difficulty separating polyps endoscopically

This method has been used in studies describing fibrin-driven matting in eosinophilic inflammation.

Interpretation of Figure 4a: Figure 4a demonstrates mild-to-moderate matting, characterized by:

  • matted clusters of small polyps
  • visible but partially fused borders
  • early fibrinous adhesion

This pattern corresponds to Stage 1 ECRSwNP in our classification.

Change in Text 4: Lines number: 183–186

Question 5: Several abbreviations appear mismatched or incorrect.

Response 5: We thank the reviewer for noting this error. The incorrect definitions in the Abbreviations section were caused by template placeholders in the MDPI manuscript format. We have now thoroughly reviewed and corrected all abbreviations to ensure accuracy and consistency.

Change in Text 5: Reviewed and Corrected.

Reviewer 2 Report

Comments and Suggestions for Authors

Hello thank you for your paper that concerns CRS with nasal polyps

You propose a classification into 3 categories:CRSwNP with eosinophils/CRS wNP witout eosinophils/ and CCAD

This is nice but we can read such classification in the last edition of the EPOS 

Clinically it is important to differentiate these 3 different entities but if you present a review of your cases you do not mention if the treatment is the same of different; What is the place for intranasal steoids:necessary for these 3 entiites?

What is the place and the result with bilogics ? for which one? outcomes?

So in fact if EPOS has already prosed this differentiation you refine the diagnostic workup including the CT and the endoscopy

basically the differences can be made based on the histopathology and the endoscopy

You must include allergic testing because CCAD is always associated to allergy

Wehn you seen polyps in the olfactory cleft, posterior septum superior meatus what is the differential diagnosis? do you consider the possible presnece of Hamartoma? ReAH? or other?

So your paper is well written but except the inclusion of the CT and the nasal endoscopy i do not find new original information concerning the last edition of EPOS

Author Response

Question: Hello thank you for your paper that concerns CRS with nasal polyps . You propose a classification into 3 categories:CRSwNP with eosinophils/CRS wNP witout eosinophils/ and CCAD. This is nice but we can read such classification in the last edition of the EPOS. Clinically it is important to differentiate these 3 different entities but if you present a review of your cases you do not mention if the treatment is the same of different; What is the place for intranasal steoids:necessary for these 3 entiites? What is the place and the result with bilogics ? for which one? outcomes? So in fact if EPOS has already prosed this differentiation you refine the diagnostic workup including the CT and the endoscopy, basically the differences can be made based on the histopathology and the endoscopy. You must include allergic testing because CCAD is always associated to allergy. Wehn you seen polyps in the olfactory cleft, posterior septum superior meatus what is the differential diagnosis? do you consider the possible presnece of Hamartoma? ReAH? or other?. So your paper is well written but except the inclusion of the CT and the nasal endoscopy i do not find new original information concerning the last edition of EPOS

Response: We sincerely thank the reviewer for the thoughtful and detailed evaluation of our work and for highlighting the relationship between our proposed classification and the most recent EPOS 2020 framework for CRS. We fully agree that EPOS has already distinguished clinically relevant entities such as eosinophilic CRSwNP, non-eosinophilic CRSwNP, and CCAD, and our intention was not to replace this classification, but to operationalise and refine it into a pragmatic, endoscopy and CT based system that can be applied in routine practice and in large-scale studies. While EPOS 2020 conceptually defines primary CRS phenotypes and locoregional variants such as CCAD, it does not provide a unified, imaging and endoscopic staging tool that can be directly used to stratify patients with CRSwNP into reproducible categories within a single cohort. In this study, we propose an integrated classification that assigns each patient with CRSwNP to one of three phenotypes (ECRSwNP, NECRSwNP, and CCAD) based solely on endoscopic findings, CT patterns, and simple laboratory parameters, and we validate these assignments in a large cohort of 2060 patients. We have now clarified this positioning with respect to EPOS in the Introduction and Discussion. 

With regard to treatment, we agree that the clinical value of distinguishing these entities depends on how they relate to therapeutic strategies, including intranasal corticosteroids and biologics. The present study was designed as a cross-sectional, diagnostic and staging project rather than as an interventional or longitudinal outcomes study, and detailed treatment response data were not systematically collected for all patients. We have added a paragraph in the Discussion to clarify that, in line with EPOS 2020, all three phenotypes are managed with saline irrigation and intranasal corticosteroids as baseline therapy, while systemic corticosteroids and biologics are currently reserved for severe, type 2 dominant ECRSwNP that remains uncontrolled despite surgery and optimal topical treatment. For NECRSwNP, which usually shows a non type 2 inflammatory profile, and for CCAD, which typically has localized disease without pansinusitis or marked osteitis, the role of biologics is much more limited under current guideline recommendations. We emphasise that our classification provides a structural framework that can be used in future prospective studies to compare treatment outcomes and responses to biologics across clearly defined phenotypes, and we have noted this explicitly as an area for further research. 

Regarding allergy, we agree that CCAD is strongly associated with allergic disease and that allergic evaluation is an essential component of the diagnostic workup. In our cohort, CCAD was diagnosed based on the typical endoscopic and CT features of central compartment involvement, together with a clinical history suggestive of allergy and, where available, elevated total IgE or positive allergy testing. We have now clarified in the Materials and Methods that allergic assessment was part of the evaluation and that CCAD should be interpreted in the context of an underlying allergic background, consistent with EPOS 2020. 

The reviewer also raises an important point regarding differential diagnosis when polyps are seen in the olfactory cleft, posterior septum, or superior meatus. We fully agree that lesions such as respiratory epithelial adenomatoid hamartoma (REAH), other hamartomatous or benign glandular lesions, and neoplastic entities need to be considered. In our study, patients with suspected neoplastic or focal benign lesions were excluded based on endoscopic appearance, CT findings, and histopathology when indicated. We have now added a statement in the Methods to make explicit that cases with REAH, inverted papilloma, other sinonasal tumours, or atypical focal lesions were not included in the CRSwNP cohort, in order to avoid misclassification.

Finally, we recognise the reviewer’s concern that, apart from the inclusion of CT and nasal endoscopy, the paper may not appear to provide sufficient novelty beyond EPOS 2020. We have therefore strengthened the Discussion to better highlight the specific contributions of our work. First, our study provides phenotype-specific endoscopic and radiologic patterns, including the description of matting, halo sign, osteitis burden, and the mulberry appearance of the inferior turbinate in advanced CCAD, each mapped to a staged severity system within a large real-world cohort. Second, we integrate these findings into a single, clinically applicable classification and staging system that can be used even in settings where extensive biomarker panels are not available, thus translating the EPOS conceptual framework into a practical diagnostic tool. Third, we identify distinct distributions of systemic markers and comorbidities across the three phenotypes, which can inform selection of candidates for biologics and guide future outcome studies. We hope that these clarifications make the added value of our work, in relation to the EPOS 2020 classification, more evident.

Change in Text: Lines number:

  • 87–96
  • 133-138
  • 150–158
  • 512–527
  • 529–538

Round 2

Reviewer 2 Report

Comments and Suggestions for Authors

The authors have made some corrections according to my previous comments.

Once again whne we read the EPOS 2020 we can find sucha classification of CRS wNP into these 3 different subtypes proposed by the authors: CRS wNP and eosinophils/ CRS w NP without eosiniophils and CCAD

The differences proposed in this paper compared to EPOS guidelines

are

first to use systematically the CT and the nasal endoscopy. In EPOS the definition of the CRS is based on the symptoms and an imaging CT or Nasal endoscopy

EPOS proposes to classifiied CRS wNP as a primary CRS diffuse bilateral type 2 because presence of eosino in the tissue

Non eosinophilic CRS wNP  is classified as CRS non type 2 without eosinophils and with a diffuse form bilateral

CCAD  is a primary CRS diffuse bilateraltype 2

In this paper the authors analysed in depth these 3 entities  and refined the classification;

When we opt for this classification allergic testing is also required because allergy is systematically associated to CCAD

In conclusion this paper does not propose something rdically different than what is written in the EPOS 2020 but insist on the role of a systematic nasal endoscopy and CT combined to a thorough medical history and histopathological evaluation